# Women’s Narratives on Infertility as a Traumatic Event: An Exploration of Emotional Processing through the Referential Activity Linguistic Program

**DOI:** 10.3390/healthcare11222919

**Published:** 2023-11-07

**Authors:** Alessia Renzi, Rachele Mariani, Fabiola Fedele, Vito Giuseppe Maniaci, Elena Petrovska, Renzo D’Amelio, Giuliana Mazzoni, Michela Di Trani

**Affiliations:** 1Department of Dynamic and Clinical Psychology and Health Studies, Sapienza University of Rome, Via degli Apuli 1, 00185 Rome, Italy; rachele.mariani@uniroma1.it (R.M.); vitogmaniaci@gmail.com (V.G.M.); giuliana.mazzoni@uniroma1.it (G.M.); michela.ditrani@uniroma1.it (M.D.T.); 2ART Italian National Register, National Centre for Diseases Prevention and Health Promotion, Italian National Health Institute, Viale Regina Elena 299, 00161 Rome, Italy; fabiola.fedele@iss.it; 3Derner School of Psychology, Adelphi University, Garden City, NY 11530-0701, USA; petre813@newschool.edu; 4Department of Gynecologic-Obstetrical and Urologic Sciences, Umberto I Hospital, Sapienza University of Rome, 00185 Rome, Italy; renzo.damelio@uniroma1.it

**Keywords:** infertility, assisted reproductive technology treatment, women, narratives, linguistic measures, trauma

## Abstract

Background: the diagnosis of infertility and its related treatment can be traumatic, leading to profound psychological distress and a variety of psychopathological symptoms. The primary objective of this study is to contrast the linguistic features of narratives from women undergoing Assisted Reproductive Treatment with those of women not undergoing any fertility treatment. This study examines the speech of both groups of individuals as an indicator of their capacity to cope with current and past distressing experiences. Method: 44 women (mean age 36.05; SD = 4.66) enrolled in a fertility medical center in Rome, and 43 control women (mean age 36.07; SD = 3.47) completed a socio-demographic questionnaire and a semi-structured interview designed to collect their memories of a neutral, a positive, and a negative event. This interview also aimed to investigate: (a) (for women with fertility difficulties) how they realized they and their partner had fertility problems and a description of an event when they talked about these difficulties with their partner; and (b) (for control group participants) the most difficult moment of their pregnancy and an event when they talked about it with their partner. The interviews were audio recorded and transcribed, and the text was analyzed using the referential process (RP) linguistic measures software. Results: Mann–Whitney non-parametric U tests for the independent samples showed several significant differences regarding the linguistic measures applied to the narratives of neutral, positive, negative, and difficult experiences in the form of a linguistic style, with more intellectualization and defenses in all the narratives associated with the women with fertility problems compared to the women in the control group. Conclusions: the traumatic and painful experience of infertility and ART seems to characterize the whole mode of narrating life experiences. Present findings sustain the importance of helping women to elaborate on their experience and to understand and recognize the difficult feelings that are activated in relation to the difficulties of having a child.

## 1. Introduction

The World Health Organization (WHO) defines infertility as the failure to achieve a clinical pregnancy after at least one year of regular, unprotected sexual intercourse [1,2], and estimates that infertility is diagnosed in about 20% of reproductive age couples in developed countries. Worldwide, infertility represents a major life crisis, threatening parenthood and life purpose and thus necessitating an emotional adjustment process [1,3] since it impedes personal, familial, and social stability [4]. Generally, experiencing complications with fertility appears to be associated with marital and sexual challenges, mood and anxiety disorders, feelings of guilt, a diminished perception of quality of life, identity issues, stigma, and social isolation [5,6,7,8,9,10,11]. It is noteworthy that these adverse consequences tend to impact women more severely and frequently than men [12,13,14,15,16]. In fact, while infertility may result from both male and female factors, prevalent social biases tend to associate infertility primarily with women. This is a potential reason why women, in particular, often carry the stigma and guilt associated with infertility and its related treatments [4,17]. Fatigue associated with efforts to conceive, uncertainty about treatment outcomes, physical pain and discomfort from medical procedures, and the economic burden all contribute to increased psychological distress [18,19,20]. Consequently, infertility and its management can be seen as a traumatic experience [4,21].

In this context, a recent study highlighted that 41.3% of women dealing with infertility exhibited symptoms consistent with posttraumatic stress disorder [4]. Consistent with research findings in the area of trauma, studies have found that reproductive problems challenge normal coping mechanisms, placing significant strain on those impacted by the experience of infertility [22,23]. In fact, women who experience reproductive difficulties often perceive theirs and their partner’s infertility as a central theme in their lives, struggling to move forward with life transitions, and continuously grieving the unfulfilled mother–child relationship [24,25,26].

One way to understand how people articulate emotionally difficult experiences, such as infertility, is through analyzing the way they speak and the language they use to describe the experiences they have faced. Indeed, language characteristics reflect the ability to connect emotions to words and thoughts [27]. In line with Bucci’s multiple code theory (MCT), that is, a psychodynamic theory of the mind allowing the consideration of phenomenon in both their physical and mental dimensions, linguistic analysis of spoken or written text enables the assessment of connections between various methods or levels of processing and elaborating information [28,29,30,31]. These methods are (i) the sub-symbolic system, which involves simultaneous analogical processing of various forms of information, playing a crucial role in recognizing non-verbal communication and organizing the affective core; (ii) the non-verbal symbolic system, which works with discrete images or representations that emerge from the continuous process of the sub-symbolic system; and (iii) the verbal symbolic system, in which images and representations can be transformed into words.

According to MCT, the three systems are interconnected through the referential process (RP), representing the process of connecting non-verbal material with material that can be communicated to others using language [27,28,29,31,32,33,34]. The RP is important because it allows individuals to communicate their emotional experience to others, while also serving as a form of self-regulation for emotions. Some experiences, such as internal conflict or experiencing a traumatic event, can cause an interruption and/or dysregulation in the RP, causing a disconnection between systems. This leads to challenges in emotional self-regulation, including difficulty in expressing emotions verbally and constructing meaning or cognitive understanding of emotional experiences [35]. Specific life experiences are consolidated into memory schemas [36], but according to Bucci, emotional experiences are also consolidated into specific memory patterns connoted by emotional experiences and visceral body patterns called emotion patterns. Just as memory patterns signify autobiographical experience, emotion patterns also signify patterns of visceral constellations and mnemonic representations that are specific to each individual [37]. In highly traumatic or conflictual life events, related emotion patterns are also more likely to generate dysregulated and difficult-to-manage arousal [31,38]. MCT represents a valid and actual theoretical framework used to explore how the words we use to construct a story related to our experience may reflect our elaboration capabilities with regard to a difficult/traumatic event.

In this context, possessing limited abilities to regulate one’s emotions could pose a risk factor, since a crucial aspect of coping with difficult/traumatic experiences involves maintaining a good connection with one’s emotions and needs [39]. Alexithymia, characterized as a disorder of emotion regulation involving challenges in recognizing and articulating emotions, utilizing imagination to manage distressing feelings, and communicating one’s needs to others to obtain social support [39,40], has been associated with infertility [21,41,42,43,44,45,46]. Furthermore, alexithymia is associated with the use of maladaptive coping strategies, which can result in greater levels of psychological symptomatology. These factors can all contribute to unsuccessful ART outcomes and/or treatment dropout [41,44,47,48].

Few studies have employed a narrative approach, representing a qualitative method used to explore how individuals make meaning of their life experiences through analyzing meanings conveyed by the content and structure of the discourse [49] in ART. Their main aim was to identify the motivation for childbearing and focus on the impact of infertility and the associated medical procedures on women. In Cipolletta and Faccio [50], interviews with women and men undergoing ART highlighted four main themes of clinical relevance that described their experience: (1) the present moment, (2) waiting, (3) hope, and (4) death. In Langher et al. [25], four principal themes were identified using emotional text analysis: (1) an inclination to self-sacrifice, seen as the price to pay for the desired success of the treatment; (2) pursuit of inclusion in the world of procreative mothers; (3) precarious equilibrium between a deep desire for a baby and withdrawal from the treatment; and (4) surrender to any possible consequence in order to obtain the desired mother–child relationship. Regarding RP in the context of ART, computerized linguistic analyses have been employed to assess expressive writing protocols from women undergoing ART procedures. These analyses have revealed that women with severe alexithymia exhibit lower scores in RP indexes. This finding suggests that alexithymia may have a detrimental impact on the symbolization process, potentially hindering the full connection between emotions and the ability to express them in words [45].

To the best of our knowledge, this is the first study aiming to investigate RP measures in narratives of women with and without fertility problems discussing their traumatic experiences associated with their parenthood journey. Specifically, we aim to compare the two groups of women based on the linguistic measures applied to their narratives regarding both common topics (respectively, emotionally neutral, positive, and negative events), and those related to specific key topics: (a) fertility problems for the group of women undergoing ART, and (b) the most difficult experience during pregnancy for the control group. Moreover, we aim to explore differences in alexithymia scores between the two groups of women to rule out the possibility that differences in RP linguistic measures may be due to greater difficulties in connecting with personal emotions and needs. We hypothesize that difficulties in conceiving may represent a traumatic event capable of causing a disconnection between sub-symbolic and symbolic systems, leading to increased dissociative processes, and resulting in RP indexes that do not align with the specific narrative topics.

## 2. Materials and Methods

### 2.1. Participants

Participants of the clinical sample (individuals with fertility problems) were recruited from women undergoing ART at a specialized hospital department in Rome, in accordance with the following inclusion criteria:-Are at their initial medical consultation before commencing a specific cycle of ART;-Are peceiving ART for a fertility problem rather than pre-implantation genetic diagnosis (PGD);-Are without biological children;-Demonstrate proficiency in understanding, speaking, and writing in Italian.

We excluded women who had reported in their anamnesis to have a previously diagnosed psychiatric disorder and/or women who had already participated in the study in a previous ART cycle. Participation in the study was proposed to the first 49 eligible women attending the medical centre, and 44 agreed to participate and completed the study’s protocol (acceptance rate 89.5%).

Participants in the control group (women without a history of fertility problems) were recruited using a convenience method (snowball procedure) and had to meet the following inclusion criteria:-Have given birth to a child in the last three years through spontaneous pregnancy;-Have no previous experience with ART;-Are married or in a cohabitating relationship;-Are capable of understanding, speaking, and writing in Italian;-Have not being diagnosed with a psychiatric condition prior to or after the pregnancy (as for the women in the fertility problems group).

Participation in this study was proposed to 45 women; 2 women were excluded due to having previous experience with ART, whereas the 43 eligible women agreed to participate and completed the protocol.

### 2.2. Measures

In the present study, all participants completed a socio-demographic questionnaire realized ad hoc to collect the information of interest for the investigation, the 20-item Toronto Alexithymia Scale (TAS-20) [51,52] was used to evaluate alexithymia levels, and a semi-structured interview was conducted to collect women’s narratives on the topics of interest. Moreover, the Italian Discourse Attribute Analysis Program (IDAAP) [53] with its dictionaries and measures validated for the Italian language was applied to the transcribed texts to investigate the RP.

Socio-demographic questionnaire. This questionnaire was designed to gather common demographic information from both groups of women, including age, social status, level of education, occupational activity, and previous miscarriages. Additionally, it included group-specific inquiries. For women with fertility issues, we collected data on previous ART attempts, the specific techniques employed, and potential causes of fertility problems. In the case of women without a history of fertility issues, we collected information concerning the age of their first pregnancy, the number of children they have, and the amount of time it took to achieve the first pregnancy.

The 20-Item Toronto Alexithymia Scale [44,51,52,54]. The TAS-20 was administered to both groups. This scale assesses three key characteristics of the phenomenon, organized into three factors: difficulty identifying feelings (DIF), difficulty describing feelings (DDF), and externally oriented thinking (EOT). Participants rated each item on a 5-point Likert scale, ranging from “1 = strongly disagree” to “5 = strongly agree”. The TAS-20 total score can vary between 20 and 100, and individual scores for each factor are also calculated. Previous studies have demonstrated satisfactory internal reliability (Cronbach’s alpha for total score = 0.75) and test–retest reliability (r = 0.83) for the TAS-20. In the present study, a Cronbach’s alpha of 0.83 was obtained for the TAS-20 total score.

Semi-structured Interview: a specific semi-structured interview was employed to collect narratives for each group of women. The interview comprises three common questions regarding the history of a neutral, positive, and negative event and two questions specific to each group that aim to explore their infertility experiences and the most difficult event faced during pregnancy. More specifically, all the participants answered the following three questions: (1) can you tell me how you spent the last weekend? (neutral event); (2) can you tell me about a positive event of your life? (positive event); (3) can you tell me about a negative event of your life? (negative event). Only the women in the group with fertility issues answered these additional questions: (1) can you tell me about when you realized you, as a couple, were experiencing difficulties with fertility?; (2) was there any moment in particular when the problem became explicit and was it clear to you?; (3) please tell me about a time when you talked to your husband/partner about problems with fertility? Only the women in the group without fertility problems answered these additional questions: (1) can you tell me about the event that you think has been the most difficult to deal with in your pregnancy since you decided to become a mother?; (2) please tell me about a time when you spoke to your husband/partner about concerns with having a child.

Computerized linguistic measures of Referential Process:-The Italian Discourse Attribute Analysis Program (IDAAP) [53] is a computer-based text analysis system developed by Maskit [55] that utilizes weighted and unweighted dictionaries and provides output based on an exponential smoothing operator. In our study, we selected specific dictionaries and derived measures that have been constructed and validated for the Italian language to investigate the RP in written text.-The Italian Weighted Referential Activity Dictionary (IWRAD) [56] is a computerized tool used to measure Referential Activity (RA) in the Italian language. A high score on this measure indicates a high level of RA, which corresponds to increased concreteness, specificity, clarity, and imagery in the speech sample. What makes the IWRAD valuable is its ability to evaluate linguistic style beyond content and to capture the implicit aspects of emotional engagement. Elevated scores on this measure help to identify and analyze the symbolizing phase of the referential process, as defined by MCT.-The Mean High–Italian Weighted Referential Activity Dictionary (MH-IWRAD) [56] is derived from IWRAD scores and serves as an indicator of the Referential Activity Intensity Index, representing a high degree of emotional engagement detected in speech. It specifically measures how elevated the IWRAD scores are when they surpass the mean. This calculation involves considering only words with IWRAD scores exceeding the neutral value and then computing the average of these IWRAD scores. Essentially, it quantifies the upward fluctuations in RA scores.-The Italian Weighted Reflection and Reorganization List (IWRRL) [56] assesses the extent to which a speaker is attempting to recognize and comprehend the emotional importance of an event or series of events in their own life, someone else’s life, or within a dream or fantasy. It does not pertain to abstract reflection, but rather focuses on a person’s reasoning related to a vividly experienced event. The IWRRL consists of weighted Italian words associated with the reorganization and reflection function. High scores on this measure indicate a high level of reflection and reorganization, aligning with the concepts of MCT.-The Mean High–Italian Weighted Reflection and Reorganization List (MH-IWRRL) [56] is derived from IWRRL scores and serves as a measure of the high intensity of the reflection and reorganization function within a person’s speech. It shows how high the IWRRL score is when it exceeds the mean value. This measure is calculated through considering only the words with IWRRL scores that surpass the neutral value, and then computing the average score for these specific words. Essentially, it quantifies the upward fluctuations in reflection and reorganization scores.-The Italian Reflection Dictionary (IRefD) comprises Italian words associated with cognitive and logical functions and communication processes that involve cognitive abilities. It quantifies abstract reflection and the act of distancing oneself from emotional experiences, calculated as the ratio of IRefD words found in speech. The IRefD measure has proven to be a reliable indicator of defensive intellectualization in studies related to psychotherapy processes [28,47].-The Italian Sensory Somatic Dictionary (ISensD) is a computerized content dictionary that focuses on the body and bodily activities. It compiles words associated with the body, physical symptoms, movements, and all aspects related to bodily experiences [28].-The Italian Affect Dictionary (AffD) is a computerized content dictionary designed to measure different aspects of affect, including negative (IAffN), positive (IAffP), and neutral (IAffZ) affect [46].

### 2.3. Procedure

Data collection for this study took place from October 2021 to May 2023. Ethical approval for the present study was granted by the Ethics Committee of the Department of Dynamic and Clinical Psychology and Health Studies “Sapienza”, University of Rome. The research was conducted in compliance with the code of ethics of the World Medical Association (Declaration of Helsinki) for experiments involving human participants.

The women in the clinical group were informed of the study during their first gynecological visit in the Hospital ART Department in Rome. The psychologist and the gynecologist screened the women for eligibility during clinical evaluation and after the medical visit, the psychologist asked the eligible women for their consent in participating in this study. Each woman participating in the study signed an informed consent form before completing the measures and the semi-structured interview. The study protocol was performed within the medical center by a qualified psychologist in a dedicated room, guaranteeing the participants’ privacy.

The women in the control group of this study were recruited using a non-discriminatory technique known as the snowball method. This method involves initially recruiting study participants who, in turn, provide referrals to new participants, creating a chain of referrals. Each participant in this chain signed an informed consent form before completing the measures and participating in the semi-structured interview. The research protocol took place in a setting agreed upon by each participant in which her privacy could be guaranteed and was implemented by a qualified psychologist. The interviews were carried out by two psychologists that were the same for both the clinical and the control groups to reduce potential bias. In the present study, each interview was recorded, transcribed verbatim, and processed using the computerized IDAAP. The IDAAP was applied to the transcribed text separately for each topic: neutral, positive, and negative events and key topic episodes (experience with fertility problems/experience with traumatic pregnancy).

### 2.4. Statistical Analyses

The statistical analyses for the present study were executed using the Statistical Package for Social Science—24 (SPSS version 24, Armonk, NY, USA). Continuous variables were described as means and standard deviations, whereas the discrete variables were reported as percentages and frequency. Using one-way ANOVAs and Chi-square tests, the homogeneity of the two groups of participants was evaluated, respectively, for the sociodemographic continuous variables (normally distributed) and discrete variables. Mann–Whitney non-parametric U tests for independent samples were employed to evaluate the differences between the two groups of women, considering that both the linguistic measures and the alexithymia values did not follow a normal distribution. A *p* value < 0.05 was considered significant.

## 3. Results

No woman in either group was excluded for psychiatric concerns; however, three women were excluded due to language issues in the clinical group (not Italian speaking).

The sociodemographic characteristics of the women are shown in Table 1. The two groups of women produced homogenous results for the variables investigated, except for the number of times they had intercourse since their pregnancy attempts, which was considerably higher in the group without fertility problems (see Table 1).

Specifically, the group of women with fertility problems had a mean age of 36.05 years (SD = 4.66), and all were married or cohabitating. Out of all participants, 6.8% of women reported having completed 8 years of education, 25% had 13 years of education, 61.4% had completed 16 years of education, and 6.8% had more than 16 years of education. Regarding employment status, 77% of the women were employed, 11.5% worked as freelancers, and 11.5% were housewives. Additionally, 20% of the participants reported having experienced a miscarriage.

The control group had a mean age of 36.07 years (SD = 3.47), and all were married or cohabitating. Of all the participants, 4.6% of the women reported having completed 8 years of education, 32.5% had 13 years of education, 51.3% had 16 years of education, and 11.6% had over 16 years of education. Regarding employment status, 67% of the women were employed, 18.5% worked as freelancers, and 14% were housewives. Additionally, 27% of the participants reported having experienced a miscarriage.

According to the Mann–Whitney non-parametric U tests for independent samples, the analysis showed several significant differences regarding linguistic measures applied to the narratives of neutral, positive, negative, and difficult experiences (see Table 2) in the form of a linguistic style, with more intellectualization and defenses in all the narratives associated with the women with fertility problems, compared to the women in the control group. More specifically, the results show that in the neutral episode, women in the group with fertility problems recounted autobiographical memories with more sensory-somatic words, while those in the control group tended to reflect and reorganize the experience, even in episodes that they define as neutral. Regarding the episodes evoked as positive, the control group presented higher levels of sensory-somatic and reflective words, words with negative affect, and higher disfluency. In contrast, with respect to the events evoked as negative, we found that the women in the control group had greater disfluency and greater symbolization. Finally, in the narrated episodes about infertility, the clinical group used more words overall, more reflexive and abstract words, and fewer sensory-somatic words and words with negative affect.

Data analysis does not reveal any significant difference between the group with regard to alexithymia total and factor score; nevertheless, a tendency to the significance (*p* = 0.06) emerged in the externally oriented thinking scores that resulted in them being higher in the group of women with fertility problems compared to the control group.

## 4. Discussion

The principal aim of this study is to investigate the RP measures applied to narratives of women with and without fertility problems who are discussing their traumatic experiences associated with their parenthood journey. We hypothesize that difficulties in conceiving may represent a traumatic event, evoking a disconnection between sub-symbolic and symbolic systems reflected in greater dissociative processes and resulting in RP indexes that are not coherent with the specific narrative topics. In the context of this hypothesis, we observed that when responding to questions regarding their awareness of infertility and the most challenging event of their pregnancy, women in the clinical groups tended to use more words. They also scored higher on the IWRRL and IRef indexes, indicating greater activation of the processes involved in trying to intellectualize, recognize, reorganize, give meaning to the emotional significance of these events. This aspect could reflect an attempt to abstract and generalize the experience so as to create emotional distance and view the event from an external and more distanced perspective. Generally, this reorganizing phase is effective after individuals have been in contact with their emotional experience. However, if this phase takes place before or instead of emotional arousal, it can function as a defense mechanism, leading to disconnection from emotional schemas. In fact, the use of abstract and reflective words is strongly associated with intellectualization and the defensive strategy of narrating a memory [57]. Thus, it seems possible to hypothesize that women with fertility problems, when discussing their difficult experiences that are still to be fully elaborated, are engaged in reflection but are also employing defense mechanisms, as indicated by the use of intellectualization as a means of distancing themselves from emotional elements. It is probably the case that women without fertility problems who are discussing difficult events from the past are abler to accede to the emotional and depressive dimensions of the experience, indicated by their talking with a less defensive style, as demonstrated by the higher scores in negative affect and sensory-somatic words. Regarding the narratives exploring a positive, negative, and neutral event, all the differences observed point towards a higher level of emotional and reflective engagement in the speech of women in the control group when compared to women facing fertility problems. The most noteworthy differences were observed in the neutral and positive episodes. In both situations, individuals in the control group connected with conflicting emotions with greater ease. Indeed, within the category for neutral events, we observed a greater use of sensory-somatic language and less intellectualization. Particularly in the narratives of a positive event of the control group, there was a higher overall level of emotional activation, increased expression of negative affect, greater use of sensory-somatic language, increased disfluency, and more frequent use of abstract words. This suggests that the control group, which engaged in sharing their life experiences with greater freedom, utilized a more nuanced and unfettered narrative style.

In contrast, when the clinical group was prompted to discuss experiences unrelated to infertility, they generally remained closely adherent to the interview context and struggled more to evoke a distinct narrative style. The traumatic experiences that the women in the clinical group are undergoing permeates through the entire interview, and also influences their memory retrieval. In other words, our initial hypothesis suggested that the recall of neutral, positive, and negative episodes might lead to a freer intermediate space. The results of this study underscored the profound emotional impact that traumatic life experiences can have on an individual [58].

More specifically, when compared to the clinical group, women in the control group reported experiencing greater emotional arousal, disfluency, and use of affective words. This suggests that the experience of infertility is so challenging for the clinical population that it hinders their use of affective and emotional words, as well as their overall engagement in speech, even in the positive and neutral narratives. Indeed, advances in reproductive medicine have created new possibilities for family building, but with these also comes an emergence of emotional and moral dilemmas. The psychological understanding of the impact of these experiences has unfortunately lagged behind these developments [59].

In this regard, although no significant difference in alexithymia scores between the groups emerged, there is a noteworthy trend towards significance in the EOT factor. This trend suggests a greater reliance on a concrete cognitive approach, possibly as a defense mechanism against the challenges of a difficult and overwhelming experience. This finding aligns with a recent study that reported similar levels of alexithymia among women facing infertility problems, compared to women without fertility problems [60], although it is not consistent with previous infertility studies [41,43,61].

Concerning the neutral event, the clinical group exhibited higher scores in the indexes, indicating the activation of a process aimed at recognizing and understanding the event. However, when asked about how they spent the weekend, the women in the clinical group provided narratives related to a period of waiting and preparation for ART, rather than describing a genuinely neutral experience. This suggests that, particularly for women undergoing ART, nothing can truly be considered neutral in their experiences.

Table 3 reports some clinical examples of women’s narratives showing different emotional processes and engagement in the narrative task according to group belonging.

### Limitations

The present findings need to be interpreted considering some limitations. Firstly, the sample size was small, which can be attributed in part to the challenges in enrolling participants for the clinical sample. Nevertheless, the sample size aligns with previous studies employing a qualitative approach [25,50]. A second limitation is the inclusion of only female participants. In future studies, it would be interesting to include male partners as well. This appears to be relevant considering that the characteristics of male partners, such as coping strategies, distress levels, and sexual concerns can also affect their female partner’s stress, quality of life, and sexual satisfaction [62,63]. Third, our choice to use partially different questions is due to the difficulty in finding a comparable experience in terms of difficulties experienced by women undergoing ART and those who experienced physiological pregnancy in their parenthood journey. Fourth, it is important to note that the recruitment procedure differed for the clinical and control groups, with clinical participants coming from a single fertility clinic and controls recruited through the snowball method. This may introduce a potential selection bias. Fifth, this study did not include an assessment of certain psychological dimensions, which should be considered in future studies.

## 5. Conclusions

This study confirms the relationship between body–mind and language. Certainly, language serves as a means to articulate the psychobiological processes within an individual, serving as a valuable indicator for assessing the depth of affective processing during an experience. Nevertheless, there is a need for more extensive documentation and research on this aspect. Expanding our knowledge in this area can provide valuable information and resources to enhance health interventions, particularly in supporting individuals undergoing invasive medical treatments like ART. The ability of individuals to effectively process information related to a traumatic event can significantly influence both their physical and mental well-being, underscoring the importance of further investigation in this field. In this study, we found that the challenges associated with becoming a mother and the repeated attempts at ART significantly shape women’s life experiences. The episodes recounted, whether positive, negative, or neutral, are influenced by the individuals’ current experience. In other words, traumatic and painful experiences do not just impact the narrated experiences related to infertility, but they also characterize the entire way of narrating life experiences. Previous studies have found that having experienced a difficult and negative emotional state affects the mode and quality of re-enactment of autobiographical memories. Indeed, we know that recollection and retelling of an autobiographical episode is conditioned by the present time and that retelling produces a transformative event in psychotherapies, but a negative experience could also produce a negative effect on past experience. Therefore, the present study shows that the difficult ART experiences faced at the time of the interviews may have conditioned the narratives of neutral, positive, and negative events. In fact, the control group shows greater emotional activation and better symbolization processes precisely in negative narratives, as evidenced by previous studies in which negative events tend to be more vivid and engaging [64], whereas for the clinical group, difficult, traumatic experiences block the activation of emotional schemas and immersion in an autobiographical narrative flow. Indeed, we know that the recollection and narration of an autobiographical episode are influenced by the present moment, and the act of retelling can lead to a transformative event in psychotherapy. However, negative experiences can also retroactively affect how past experiences are viewed. In this context, experiencing a painful present situation and attempting to defend oneself against it can overshadow and influence all other life experiences. It is crucial to acknowledge that the emotional processing of an individual is shaped by their present experiences, current emotions, cognitive appraisals, coping efforts, and personality traits. Furthermore, one’s recollection of past emotions is not a neutral process; it can be influenced by these present factors and may carry biases into how they remember and interpret past emotional experiences. People’s memory of emotions provides succinct and readily accessible reference points that inform their understanding of how past experiences relate to their to current goals [58,65,66,67]. In addition, some research has shown that difficult moments in the present affect the quality of positive life narratives more than neutral and negative ones [66,68], which is something that our results also support. In fact, the account of positive episodes is the one category that showed a greater difference in narrative style and quality between the two groups.

This imposes a strong need to approach the process of ART with a psychological focus, especially in situations that do not result in a positive outcome. In fact, a positive outcome is one that presents the opportunity for transformation, while a negative outcome risks crystallizing a woman’s defensive mechanisms, leading to significant mental health consequences [69,70,71].

This study has relevant clinical implications, emphasizing the importance of addressing the internal emotional process experienced by women at the outset of ART interventions [25,72]. This early focus can assist women in processing their experiences and understanding and acknowledging the challenging emotions that arise in response to the uncertainty surrounding the procedure’s outcome, specifically the possibility of having a child [21]. Considering these results, it is important to note that assessing emotional defenses and implementing psychological interventions that facilitate the expression of emotional experiences could support women in staying connected with their challenging emotions. This, in turn, may enable them to reflect on and reorganize their experiences more effectively. Moreover, fostering a strong emotional connection can positively impact the outcomes of ART, as it is associated with reduced distress and increased treatment success [73].

## Figures and Tables

**Table 1 healthcare-11-02919-t001:** Sociodemographic Variables.

Sociodemographic Variables	ART Women Group (*n* = 44)		Control Women Group (*n* = 43)		F	*p*
	M	SD	M	SD		
**Age**	36.05	4.66	36.07	3.47	0.001	0.980
**Partner’s Age**	38.60	6.17	39.29	5.21	0.307	0.581
**Number of times they had intercourse since the start of pregnancy attempts (months)**	29.79	19.11	9.07	17.55	26.413	0.000
**First Pregnancy Age**			30.79	3.56		
	**N (%)**		**N (%)**		**X^2^**	* **p** *
**Previous ART attempts**						
One	27 (31%)					
Two or more	15 (12.6%)					
**Infertility causes**						
Female	12 (28.5%)					
Male	12 (28.5%)					
Both partners	8 (19%)					
Unknown	10 (23.7%)					
**Occupation**					1.169	0.557
Employee	34 (77.27%)		29 (67.44%)			
Freelance	5 (11.36%)		8 (18.60%)			
Housewife	5 (11.36%)		6 (13.95%)			
**Level of Education**					1.559	0.669
8 years	3 (6.82%)		2 (4.65%)			
13 years	11 (25.00%)		14 (32.56%)			
16 years and above	30 (68.18%)		27 (62.79%)			
**Previous Miscarriages**	9 (20.45%)		12 (27.90%)		0.660	0.417

**Table 2 healthcare-11-02919-t002:** Significant differences in linguistic measures between the two groups.

	ART Women Group (*n* = 44)			Control Group (*n* = 43)			Mann–Whitney U Test	*p*
	M	SD	M-Rank	M	SD	M-Rank		
**Difficulties in parenthood journey**								
Words	654.41	330.63	54.64	385.31	197.17	31.83	1414.000	0.000
INAffD	0.014	0.008	36.98	0.019	0.010	50.33	637.000	0.013
IRefD	0.031	0.010	55.58	0.021	0.010	30.85	1455.500	0.000
ISensD	0.039	0.013	31.62	0.0563	0.017	55.94	401.500	0.000
IWRRL	0.543	0.004	51.64	0.540	0.005	34.98	1282.00	0.002
MH-IWRRL	0.043	0.004	51.64	0.040	0.005	34.98	1282.000	0.002
**Neutral episode**								
ISensD	0.036	0.023	37.39	0.047	0.023	50.77	655.000	0.013
IWRRL	0.538	0.006	49.75	0.535	0.007	38.12	1199.000	0.032
MH-IWRLL	0.038	0.006	49.75	0.035	0.007	38.12	1199.000	0.032
**Positive episode**								
IDFD	0.050	0.029	37.24	0.067	0.031	50.92	648.500	0.012
IRefD	0.022	0.013	37.07	0.030	0.014	51.09	641.000	0.010
ISensD	0.040	0.022	34.35	0.060	0.024	53.87	521.500	0.000
**Negative episode**								
IDFD	0.051	0.027	38.68	0.064	0.033	49.44	712.000	0.047
IWRAD	0.4929	0.006	38.75	0.4997	0.006	49.37	715.000	0.049

Note: IDFD = Italian Disfluency Dictionary; INAffD = Italian Negative Affect Dictionary; IRefD = Italian Reflection Dictionary; ISensD = Italian Sensory Somatic Dictionary; MH-IWRLL = Mean High–Italian Weighted Reflection and Reorganization List; IWRAD = Italian Weighted Referential Activity Dictionary; IWRRL = Italian Weighted Reflection and Reorganization List.

**Table 3 healthcare-11-02919-t003:** Clinical vignettes of narratives produced by women.

ART Group Woman	Control Group Woman
**Positive episode**	
A positive story is when I finally decided to get engaged to my husband, to my current husband.	On the other hand, a positive story was the birth of my children. I could say, instead, that I have also experienced that in two, two completely different ways, because my first child was born at full term. Ehmm/Already pretty big, then I immediately took it home, with the cut, that is with a natural birth. Now the little one instead, ehmm I haven’t seen for 24 or 48 h, but as soon as she saw me she recognized me. So that it was a beautiful thing/a little girl who immediately recognizes her mother was a joy for me. A beautiful thing, indeed.
**Difficulties in parenthood journey** (infertility awareness process/most difficult pregnancy experience)	
When with my husband’s last spermiogram, his andrologist really told us that it was unfortunate that a “natural” pregnancy could not occur. We have been trying for two and a half years. I knew that I have polycystic ovaries so I thought it was more of a problem on my part, but then, in talking to my gynecologist and doing various checks, we learned that it wasn’t my problem; then we went to the andrologist and so on Um/last year or so…At that moment, “the world fell on me a bit” because I was hoping so much for a pregnancy of course, I hoped I wouldn’t go by the way um/mmh Medical yes/instead then also talking to my husband/we talked about it, he was close to me/he told me unfortunately we are not the only ones, we are not the only couple who have this problem but if we really want to have a child we have to resort to these medical avenues Ehm/at that moment, when I spoke to the andrologist, a lot of sadness, but then talking to my husband, it relieved me a lot… Yes, yes yes, both my family and my husband’s family and um and then a cousin of mine who is the person I always talked to about everything Yes yes very, very much Yes, for me yes a lot.	Then with the first child, I had toxoplasmosis problems during pregnancy. It was a cold shower from the start, though it was an infection, then back and all, but/the fact that there was even a slight chance that the baby might have it and might not be well I brought it along for everyone and nine months and later even when ummm/he had to follow up for a year to check he was still negative for infections. And it wasn’t a good/I know- especially as a first pregnancy, I didn’t live it light-heartedly. With the second one, I had the problem of placenta previa, so I was anxious until the thirty-second week, mm because we didn’t know if I could risk a hysterectomy or possibly just a hemorrhage, a transfusion, that the baby might not be well, might not grow. So even a C-section that however I am/I’m scared of the operating room, I’m easily impressed, so. Yes Yes Yes. Both particular, both particular. Everything went well on both sides fortunately, that’s what counts. Toxoplasmosis at six weeks pregnant, I first discovered; I was diagnosed for the first time with placenta previa on the fourteenth, i.e., low placental insertion at the fourteenth week of pregnancy was diagnosed for the first time. Yes, both. I did it, yes yes. No, the important thing was the child themselves/with XXX, the first one who was well. Well, but I immediately understood that he hadn’t contracted the infection when I saw him because he was born, although they had told me he could be blind, deaf, as soon as I saw him, I saw that he was already born with eyes open; a child already awake, that is, it could not have been otherwise. And the second instead/in any case, even if she was born a couple of days preterm for this planned cesarean, in any case a/she was born with a good weight/not/In the end, yes, everything went well.

## Data Availability

The data that support the findings of this study are available upon request from the corresponding author.

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
