# Peer review of "Women’s Narratives on Infertility as a Traumatic Event: An Exploration of Emotional Processing through the Referential Activity Linguistic Program"

_healthcare, 2023, doi:10.3390/healthcare11222919_

Round 1

Reviewer 1 Report

Comments and Suggestions for Authors

Situations of recurrent infertility have an important emotional factor, which has empirically been shown to be relevant, but more and better studies are required to highlight this relationship.

Therefore, studies like this one are necessary and relevant to the scientific community. Furthermore, this work includes the technical and methodological indications necessary to present the quality necessary to be published in this journal.

The only thing missing is a larger sample, which is why it is recommended to expand the study by incorporating a larger number of women.

Author Response

Reviewer 1

Situations of recurrent infertility have an important emotional factor, which has empirically been shown to be relevant, but more and better studies are required to highlight this relationship.

Therefore, studies like this one are necessary and relevant to the scientific community. Furthermore, this work includes the technical and methodological indications necessary to present the quality necessary to be published in this journal.

Authors: we thank the reviewer for the general positive evaluation and for the strengths highlighted.

The only thing missing is a larger sample, which is why it is recommended to expand the study by incorporating a larger number of women.

Authors: We agree that a greater sample size would be useful, and we aimed in future to realized study with a larger number of participants. The limited sample size was mainly due to the challenges in enrolling women during ART treatment. We have highlighted this point in the limits section.

“the sample size was small, which can be attributed in part to the challenges in enrolling participants for the clinical sample. Nevertheless, the sample size aligns with previous studies employing a qualitative approach”

Reviewer 2 Report

Comments and Suggestions for Authors

Concerning Italian women’s narratives on infertility as a traumatic event, this study compares the linguistic measures applied to emotionally neutral, positive, and negative events, as well as those related to: a) infertility conditions, for women with fertility problems, and b) to the most difficult experience during pregnancy, for the control group through differences in alexithymia scores. It confirms that difficulties in conceiving are capable of causing a disconnection between the sub-symbolic and symbolic systems, leading to increased dissociative processes, and resulting in Referential Process indexes that do not correspond with the specific narrative topics—results that are relevant for clinical implications.

The paper is well written, has a clear hypothesis, and is logically organized. Its major problem is that of the 61 references cited by the authors, only 19 were published in the last five years. This is too few for a scientific article. In the line by line suggested edits, those references that are particularly problematic in being more than 10 years out of date are noted (unless they are supported by current references). The authors are expected to find current references from peer reviewed journals to replace these outdated references.

Please also redo all the citations and references in MDPI style as found in the Instructions for Authors on the Healthcare website.

Line by line suggested edits.

50 None of the cited articles here are current. Here is a Google Scholar search of “infertility affects women more than men” published since 2019. There are many articles that have been returned. The authors are asked to choose from the returns of this search for their citations: https://scholar.google.ca/scholar?hl=en&as_sdt=0%2C5&as_ylo=2019&q=infertility+affects+women+more+than+men&btnG=

56 The reference associated with citation 16 is out of date. Here is a Google Scholar search of “psychological distress from Assisted reproductive technology in women” post-2019. Please choose a reference from these returns: https://scholar.google.ca/scholar?hl=en&as_sdt=0%2C5&as_ylo=2019&q=psychological+distress+from+Assisted+reproductive+technology+in+women&btnG=

65 Citation 20 is to an out of date reference. Please choose and article to reference from the returns of this Google Scholar search of “couple's infertility continuously grieving unfulfilled mother-child relationship”: https://scholar.google.ca/scholar?hl=en&as_sdt=0%2C5&as_ylo=2019&q=couple%27s+infertility+continuously+grieving+unfulfilled+mother-child+relationship&btnG=

67 Please provide an explanation (supported by current peer reviewed references) of the relationship between narratives and language characteristics—this is not self-evident. 

71 As the references related to citations 22 and 23 are more than ten years old, the authors must demonstrate that MCT is still relevant in 2023. Here is a Google Scholar search of “Bucci's Multiple Code Theory”. Please choose a current reference to demonstrate its continued relevance from this list: https://scholar.google.ca/scholar?hl=en&as_sdt=0%2C5&as_ylo=2019&q=Bucci%27s+Multiple+Code+Theory&btnG=

79 According to this Google Scholar search of “Referential Process of Bucci's Multiple Code Theory”, Bucci has authored three different papers on this topic since 2019. Please reference these rather than the those that are outdated: https://scholar.google.ca/scholar?hl=en&as_sdt=0%2C5&as_ylo=2019&q=Referential+Process+of+Bucci%27s+Multiple+Code+Theory&btnG=

86-92 As both citation 27 and 29 are to out of date references, please see if the three references to post-2019 publications by Bucci can replace the references associated with citations 27 and 29.

94-95 Please cite a current reference to support this claim.

98 Reference 30 is out of date. Please substitute it with one of the articles returned in a Google Scholar search of “Alexithymia”: https://scholar.google.ca/scholar?hl=en&as_sdt=0%2C5&as_ylo=2019&q=Alexithymia&btnG=

103 Please describe what is meant by “a narrative approach” and provide current peer reviewed references to support the description.

105-107 The study by Langher et al. supersedes that of Cipolletta and Faccio. If the authors want to use these results that are 10 years old from these authors, they need to explain that these themes still hold today. Here is a recent  publication by these same authors that may be helpful in this regard.

Faccio, E.; Iudici, A.; Cipolletta, S. To tell or not to tell? Parents’ reluctance to talking about conceiving their children using medically assisted reproduction. Sexuality Culture 201923, 525-543. https://doi.org/10.1007/s12119-019-09586-7

164-167 Were any women in the proposed group excluded for having a psychiatric condition? Please reply to this in the text.

185 A Google Scholar search of “The 20-Item Toronto Alexithymia Scale Assisted Reproductive Technology” post-2019 produced 149 returns. Please add some of these to your references cited to demonstrate that the scale is still in use today in this regard: https://scholar.google.ca/scholar?hl=en&as_sdt=0%2C5&as_ylo=2019&q=The+20-Item+Toronto+Alexithymia+Scale+Assisted+Reproductive+Technology&btnG=

265 Please explain in the text why the two periods when the data collection took place were so far apart.

292 Please provide a current reference for a similar study using SPSS version 24. As well, explain why this package was chosen for the analysis.

306-307 Table 1: Please entre the F and P columns under each heading. The data regarding the Previous ART attempts and Infertility causes should be under the ART women group, not the Control women group

323-324 Table 2: Please move the right-most column to be centered on ”P”.

459 In citing these older works, the authors also need current research to demonstrate these claims are still supported by research. The following article may be relevant in this regard.

Montijn, N.D.; Gerritsen, L.; Engelhard, I.M. (2021). Forgetting the future: Emotion improves memory for imagined future events in healthy individuals but not individuals with anxiety. Psychol Sci 202132, 587-597. https://doi.org/10.1177/0956797620972491

459-460 To make this claim, the authors need to demonstrate that the research cited from 2001 was not merely anomalous. Please find a current reference that also demonstrated that difficult moments in the present affect the quality of positive life narratives more than neutral and negative ones.

470 Citation 60 is to an older article and requires current research to support the claim. A citation to 40 may be relevant here. If not, please find a supporting current reference.

Author Response

Reviewer 2

Concerning Italian women’s narratives on infertility as a traumatic event, this study compares the linguistic measures applied to emotionally neutral, positive, and negative events, as well as those related to: a) infertility conditions, for women with fertility problems, and b) to the most difficult experience during pregnancy, for the control group through differences in alexithymia scores. It confirms that difficulties in conceiving are capable of causing a disconnection between the sub-symbolic and symbolic systems, leading to increased dissociative processes, and resulting in Referential Process indexes that do not correspond with the specific narrative topics—results that are relevant for clinical implications.

The paper is well written, has a clear hypothesis, and is logically organized. Its major problem is that of the 61 references cited by the authors, only 19 were published in the last five years. This is too few for a scientific article. In the line by line suggested edits, those references that are particularly problematic in being more than 10 years out of date are noted (unless they are supported by current references). The authors are expected to find current references from peer reviewed journals to replace these outdated references.

Authors: we thank the reviewer for the general positive evaluation and for the efforts to sustain the revision process related to an update of the bibliography providing some recent papers. We would like to specify that the use of 10 years old, or more, references are partially due to the need to cite the cornerstones of the theory and measures used (see MCT, Referenctial Acitivity dictionaries etc) and partially due to the positive evaluation of the study to sustain our hypothesis and findings thus we do not think useful to remove them. Nevertheless, we strongly considered this request and un update of the bibliography including more recent studies was realized. The new references were included in the list using a red font.

Please also redo all the citations and references in MDPI style as found in the Instructions for Authors on the Healthcare website.

Authors: the citations and references have been modified according to this suggestion through Zotero programme.

Line by line suggested edits.

50 None of the cited articles here are current. Here is a Google Scholar search of “infertility affects women more than men” published since 2019. There are many articles that have been returned. The authors are asked to choose from the returns of this search for their citations: https://scholar.google.ca/scholar?hl=en&as_sdt=0%2C5&as_ylo=2019&q=infertility+affects+women+more+than+men&btnG=

Authors: according to reviewer’s request we added the following recent manuscripts:

Hanna E, Gough B. The social construction of male infertility: a qualitative questionnaire study of men with a male factor infertility diagnosis. Sociology Health & Illness. 2020 Mar;42(3):465–80.

Turner KA, Rambhatla A, Schon S, Agarwal A, Krawetz SA, Dupree JM, et al. Male Infertility is a Women’s Health Issue—Research and Clinical Evaluation of Male Infertility Is Needed. Cells. 2020 Apr 16;9(4):990.

56 The reference associated with citation 16 is out of date. Here is a Google Scholar search of “psychological distress from Assisted reproductive technology in women” post-2019. Please choose a reference from these returns: https://scholar.google.ca/scholar?hl=en&as_sdt=0%2C5&as_ylo=2019&q=psychological+distress+from+Assisted+reproductive+technology+in+women&btnG=

Authors: according to reviewer’s request we added the following recent manuscripts:

Molgora S, Fenaroli V, Acquati C, De Donno A, Baldini MP, Saita E. Examining the Role of Dyadic Coping on the Marital Adjustment of Couples Undergoing Assisted Reproductive Technology (ART). Front Psychol. 2019 Mar 8;10:415.

Simionescu G, Doroftei B, Maftei R, Obreja BE, Anton E, Grab D, et al. The complex relationship between infertility and psychological distress (Review). Exp Ther Med. 2021 Feb 1;21(4):306.

65 Citation 20 is to an out of date reference. Please choose and article to reference from the returns of this Google Scholar search of “couple's infertility continuously grieving unfulfilled mother-child relationship”: https://scholar.google.ca/scholar?hl=en&as_sdt=0%2C5&as_ylo=2019&q=couple%27s+infertility+continuously+grieving+unfulfilled+mother-child+relationship&btnG=

Authors: according to reviewer’s request we added the following recent manuscripts:

Langher V, Fedele F, Caputo A, Marchini F, Aragona C. Extreme desire for motherhood: Analysis of narratives from women undergoing Assisted Reproductive Technology (ART). Eur J Psychol. 2019 Jun 7;15(2):292–311.

Steck B. Self Development and Parenthood. In: Adoption as a Lifelong Process [Internet]. Cham: Springer International Publishing; 2023 [cited 2023 Oct 18]. p. 7–35. Available from: https://link.springer.com/10.1007/978-3-031-33038-4_2

67 Please provide an explanation (supported by current peer reviewed references) of the relationship between narratives and language characteristics—this is not self-evident.

Authors: We try to interpret the reviewer’s comment, who ask to clarify the relations between narratives and language characteristics. We recognize that the use of the word “narrative” can be misleading, as it refers to the way in which people construct stories and the meaning they take on in relationships with others, while we intended to refer that the way in which people speak can be analysed through specific linguistic measures (referring to multiple code theory). The sentence has been modified to be more clear.

“One way to understand how people articulate emotionally difficult experiences, such as infertility, is through analyzing the way they speak, the language people use to tell the experiences faced”

71 As the references related to citations 22 and 23 are more than ten years old, the authors must demonstrate that MCT is still relevant in 2023. Here is a Google Scholar search of “Bucci's Multiple Code Theory”. Please choose a current reference to demonstrate its continued relevance from this list: https://scholar.google.ca/scholar?hl=en&as_sdt=0%2C5&as_ylo=2019&q=Bucci%27s+Multiple+Code+Theory&btnG=

Authors: according to this suggestion we added the following recent manuscripts:

Bucci W. Overview of the Referential Process: The Operation of Language Within and Between People. J Psycholinguist Res. 2021 Feb;50(1):3–15.

Fortunato A, Renzi A, Andreassi S, Maniaci VG, Franchini C, Morelli M, et al. Computerized linguistic analysis of counselors’ clinical notes in a university counseling center: Which associations correspond with students’ symptom reduction in a brief psychodynamic intervention? Psychoanalytic Psychology 2023. http://doi.apa.org/getdoi.cfm?doi=10.1037/pap0000465

79 According to this Google Scholar search of “Referential Process of Bucci's Multiple Code Theory”, Bucci has authored three different papers on this topic since 2019. Please reference these rather than the those that are outdated: https://scholar.google.ca/scholar?hl=en&as_sdt=0%2C5&as_ylo=2019&q=Referential+Process+of+Bucci%27s+Multiple+Code+Theory&btnG=

Authors: according to reviewer’s request we added the following recent manuscripts:

Bucci W. Overview of the Referential Process: The Operation of Language Within and Between People. J Psycholinguist Res. 2021 Feb;50(1):3–15.

Bucci W. Development and validation of measures of referential activity. J Psycholinguist Res. 2021 Feb;50(1):17–27.

86-92 As both citation 27 and 29 are to out of date references, please see if the three references to post-2019 publications by Bucci can replace the references associated with citations 27 and 29.

Authors: according to reviewer’s request we added the following recent manuscript:

Bucci W. Overview of the Referential Process: The Operation of Language Within and Between People. J Psycholinguist Res. 2021 Feb;50(1):3–15.

94-95 Please cite a current reference to support this claim.

Authors: according to reviewer’s request we added the following recent manuscript:

Luminet O, Nielson KA, Ridout N. Having no words for feelings: alexithymia as a fundamental personality dimension at the interface of cognition and emotion. Cognition and Emotion. 2021 Apr 3;35(3):435–48.

98 Reference 30 is out of date. Please substitute it with one of the articles returned in a Google Scholar search of “Alexithymia”: https://scholar.google.ca/scholar?hl=en&as_sdt=0%2C5&as_ylo=2019&q=Alexithymia&btnG=

Authors: according to reviewer’s request we added the following recent manuscript:

Luminet O, Nielson KA, Ridout N. Having no words for feelings: alexithymia as a fundamental personality dimension at the interface of cognition and emotion. Cognition and Emotion. 2021 Apr 3;35(3):435–48.

103 Please describe what is meant by “a narrative approach” and provide current peer reviewed references to support the description.

Authors:  we thank the reviewer for this comments that allow us to cleared this point.

Narrative analysis or approach is a qualitative method considering each individual as the element of analysis, exploring how individuals make meaning of their life experiences. According to Josselson & Hammack (2021) it represents an effort to stimulate new investigators or seasoned investigators who want to stretch their methodological expertise to consider narrative analysis. Narrative analysis interprets richly detailed life stories obtained from interviews or written documents. It goes beyond description of the text to analyse meanings conveyed by the content and structure of the discourse and always contextualizes the participant in social and historical terms. Narrative analysis allows for a careful, systematic review of meanings embedded in the language of the text and carefully tied to an analytic framework.

We added this sentence and related citation.

Few studies have employed a narrative approach, representing a qualitative method exploring how individuals make meaning of their life experiences analyzing meanings conveyed by the content and structure of the discourse (48) in ART. Their main aim was of identifying motivation for childbearing and focusing on the impact of infertility and its medical procedure on women.

105-107 The study by Langher et al. supersedes that of Cipolletta and Faccio. If the authors want to use these results that are 10 years old from these authors, they need to explain that these themes still hold today. Here is a recent  publication by these same authors that may be helpful in this regard.

Faccio, E.; Iudici, A.; Cipolletta, S. To tell or not to tell? Parents’ reluctance to talking about conceiving their children using medically assisted reproduction. Sexuality Culture 2019, 23, 525-543. https://doi.org/10.1007/s12119-019-09586-7

Authors: According to the reduce literature on this theme using a narrative approach we decided to include Cipolletta and Faccio’s manuscript also if ten years have passed considering their findings still relevant in accordance with our clinical experience.

“In Cipolletta and Faccio (49), interviews of women and men during ART highlighted four main themes, of clinical relevance, describing their experience: 1) the present moment, 2) waiting, 3) hope, and 4) death.”

The more recent papers suggested is not focused on the thematic we addressed but to a near one. We thank the reviewer because we do not know this study and we think it’s interesting and probably we will use it in other papers.

164-167 Were any women in the proposed group excluded for having a psychiatric condition? Please reply to this in the text.

Authors: no women in the control group have been excluded for psychiatric condition and a sentence on it has been added in the text.

185 A Google Scholar search of “The 20-Item Toronto Alexithymia Scale Assisted Reproductive Technology” post-2019 produced 149 returns. Please add some of these to your references cited to demonstrate that the scale is still in use today in this regard: https://scholar.google.ca/scholar?hl=en&as_sdt=0%2C5&as_ylo=2019&q=The+20-Item+Toronto+Alexithymia+Scale+Assisted+Reproductive+Technology&btnG=

Authors: we are not sure that the measure section is the right point to insert citation to support the use of the most worldwide employed instrument to assess alexithymia construct. Nevertheless, we added two citations already present in the manuscript to this point.

265 Please explain in the text why the two periods when the data collection took place were so far apart.

Authors: we thank the reviewer for the possibility to clarify this point. There were not two separate moments for data collection but it started from October 2021 to end in May 2023. If this period seems long it was partially due to the fact that data collection was slowed down since one of the psychologist responsible for the protocol implementation was in maternity period but we do not think relevant to include this information.

292 Please provide a current reference for a similar study using SPSS version 24. As well, explain why this package was chosen for the analysis.

Authors: we used this spss version because is the one available through our institution, however considering that we have performed only basic data analyses we do not think that more updated versions will have given different findings or possibilities. We do not think essential to report this information in the text.

306-307 Table 1: Please entre the F and P columns under each heading. The data regarding the Previous ART attempts and Infertility causes should be under the ART women group, not the Control women group

Authors: we thank the reviewer for this request. We have modified the table according to the suggestion.

323-324 Table 2: Please move the right-most column to be centered on ”P”.

Authors: We have modified the table according to the suggestion.

459 In citing these older works, the authors also need current research to demonstrate these claims are still supported by research. The following article may be relevant in this regard.

Montijn, N.D.; Gerritsen, L.; Engelhard, I.M. (2021). Forgetting the future: Emotion improves memory for imagined future events in healthy individuals but not individuals with anxiety. Psychol Sci 2021, 32, 587-597. https://doi.org/10.1177/0956797620972491

Authors: as suggested we added the following manuscript

459-460 To make this claim, the authors need to demonstrate that the research cited from 2001 was not merely anomalous. Please find a current reference that also demonstrated that difficult moments in the present affect the quality of positive life narratives more than neutral and negative ones.

Authors: according to this request we added the following recent citation that we think may support this point.

Williams SE, Ford JH, Kensinger EA. The power of negative and positive episodic memories. Cogn Affect Behav Neurosci. 2022 Oct;22(5):869–903.

470 Citation 60 is to an older article and requires current research to support the claim. A citation to 40 may be relevant here. If not, please find a supporting current reference.

Authors: according to reviewer’s suggestion the reference n.40 in the not revised manuscript that is langher et al. 2019 has been added to the .n60.

Reviewer 3 Report

Comments and Suggestions for Authors

 Abstract:

Introduction: This study examines the speech of both groups of these individuals as an indicator of their capacity to cope with current and past distressing experiences. Were these two groups of women?

Were the first group of women infertile and the second group (control) of fertile women????

What was the study method? Qualitative? Quantitative? Or mix method?

Why were the interviews not coded and themes extracted to be compared between groups?

How has Mann Whitney been used?

Introduction is too long, it should be shortened and it should focus on the purpose of the study. Although the purpose of study is not clear.

Method:

Why were these numbers included in each group?  ‘Participation in the study was proposed to 49 eligible women and 44 agreed to participate (acceptance)”?

What was the sample size formula?

No woman was excluded for psychiatric concerns, whereas three were excluded for language issues (not Italian speaking).

This sentence should be moved to the result section.

What was the population research? How was the sampling method? Purposive? Random? Convenient?

From line 148 to 155, it should be transferred to the result section.

The authors wrote that we excluded women that reported being diagnosed with a 164 psychiatric condition prior to or after their pregnancy.

From line 165 to line 174, it should be transferred to the results section.

Please specify first in the measure section what questionnaires were used in this study?

Line 209 and 210 are duplicates.

Data must be organized and interpreted correctly. How many sections did the linguistic measures section have? Are these parts measured by different parameters?

Through the one-way ANOVAs, the homogeneity of the two groups of participants was evaluated, specifically, for the sociodemographic continuous variables (normally distributed) and 296 discrete variables?

Why is the homogeneity of the groups checked?

What does table number three indicate?

  This report does not have the structure of a clinical article and cannot be followed due to clutter.

Author Response

Reviewer 3

Introduction: This study examines the speech of both groups of these individuals as an indicator of their capacity to cope with current and past distressing experiences. Were these two groups of women? Were the first group of women infertile and the second group (control) of fertile women???? What was the study method? Qualitative? Quantitative? Or mix method?

Authors: we hope the reviewer will find the answers to all these points reading the revised version of the manuscript. In detail, we aimed to compare two groups of women one facing fertility problems (since we cannot assume that is the woman to be infertile, maybe can be the partner or both) and a fertile one on specific linguistic characteristics applied to autobiographical memories related to both common topics (respectively, emotionally neutral, positive, and negative events), and to specific topics: a) infertility conditions for the group of women facing fertility problems, and b) to the most difficult experience during pregnancy, for the control group. Moreover, we aim to explore differences in alexithymia scores between the two groups of women to rule out the possibility that the differences in the RP linguistic measures may be due to greater difficulties in connecting with own emotions and needs since it is recognized that emotional difficulties can be reflected in an emotionally poor language and in a cognitive orientated narrative style. We adopted a mixed method, qualitative as regards the text produced by women but quantitative as regards the analyses realized that contemplated the use of validated quantitative programmes as those regarding the MCT dictionaries.

The aims section was modified to be more clear for readers

To the best of our knowledge, this is the first study aiming to investigate the RP measures on narratives of women with and without fertility problems discussing their traumatic experiences associated with their parenthood journey. Specifically, we aim to compare the two groups of women on the linguistic measures applied to narratives regarding both common topics (respectively, emotionally neutral, positive, and negative events), and those related to specific-key topics: a) fertility problems conditions, for the group of ART of women, and b) to the most difficult experience during pregnancy, for the control group. Moreover, we aim to explore differences in alexithymia scores between the two groups of women to rule out the possibility that the differences in the RP linguistic measures may be due to greater difficulties in connecting with own emotions and needs.

Why were the interviews not coded and themes extracted to be compared between groups?

Authors: we thank for this question. We do not code and extracted themes to compared them between groups because it was not our purpose, but may be realized in future investigations within a different theoretical framework.

How has Mann Whitney been used?

Authors: the text were processed through a computerized linguistic programme that produced scores for each dictionary employed to analyse written text (see measures section). Since this values were not normally distributed we cannot use parametric statistics (anova or t test) but we have to use non-parametric statistics as Mann-whithney U having only two groups to compare.

We modified as follows the related sentence.

Mann–Whitney U non-parametric tests for independent samples was employed to evaluate differences between the two groups of women considering that both the linguistic measures and the alexithymia values did not follow a normal distribution.

Introduction is too long, it should be shortened and it should focus on the purpose of the study. Although the purpose of study is not clear.

Authors: the introduction as well as the entire manuscript have been updated in accordance to all reviewers’ suggestions. The introduction seems focused on the addressed topics whereas the purpose has been reworded to be clearer.

Method:

Why were these numbers included in each group?  ‘Participation in the study was proposed to 49 eligible women and 44 agreed to participate (acceptance)”?  What was the sample size formula?

Authors: Thank you for this comment. As highlight by reviewer 1 the sample size is small, this was mainly due to the difficulties in recruiting participants in the clinical group for time constraints associated to the medical procedure. Moreover, it should be considered that in Italy there is still a great stigma associated to the use of ART services that couples face as a secret often no shared with family or friend. Furthermore, psychology is not a professional included in the ART team so to accede in these department to propose the participation in the study is not easy. We do not have realized an a priori-power analysis to establish the number of participants but we proposed the study to all the clinical women we have the opportunity to meet and we enrolled a similar number for control women.

No woman was excluded for psychiatric concerns, whereas three were excluded for language issues (not Italian speaking). This sentence should be moved to the result section.

Authors: This sentence has been moved as requested.

What was the population research? How was the sampling method? Purposive? Random? Convenient?

Authors: we adopted different method for the clinical and the control groups. We specified this information in the dedicated section

Participation in the study was proposed to the first 49 eligible women attending the medical centre and 44 agreed to participate (acceptance rate 89.5%).

Participants in the control group (women without a history of fertility problems) were recruited using a convenience method (snowball procedure) and had to meet the following inclusion criteria:…

From line 148 to 155, it should be transferred to the result section.

Authors: This paragraph has been moved as requested.

The authors wrote that we excluded women that reported being diagnosed with a 164 psychiatric condition prior to or after their pregnancy.

Authors: we clarified this sentence including this point in the inclusion criteria as follows

- not being diagnosed with a psychiatric condition prior to or after the pregnancy (as for the women in the fertility problems group).

From line 165 to line 174, it should be transferred to the results section.

 Authors: This paragraph has been moved as requested

Please specify first in the measure section what questionnaires were used in this study?

Authors: Hoping of having correctly interpreted this request we have inserted a brief paragraph at the beginning of the measures section to report all the measured that were used and we moved the interview section before the text illustrating the IDAAP programme.

The new brief paragraph included is:

2.2 Measures

In the present study all participants completed a socio-demographic questionnaire realized ad-hoc to collect the information of interest for the investigation, the 20-Item Toronto Alexithymia Scale (TAS-20) (50,51) to evaluate alexithymia levels, a semi-structured interview to collect women’s narratives on the topics of interest. Moreover, the Italian Discourse Attribute Analysis Program (IDAAP) (52) with its dic-tionaries and measures validated for the Italian language were applied to transcribed text to investigate RP.

Line 209 and 210 are duplicates.

 Authors: Thanks to this comment the typo has been removed.

Data must be organized and interpreted correctly. How many sections did the linguistic measures section have? Are these parts measured by different parameters?

Authors: we thank the reviewer for this comments, the IDDAP programme and related dictionaries have been applied to the entire text divided for topic. Thus they have been applied to text exploring neutral positive negative and key topic event. The dictionaries and the associated measures are those reported in the measures section. This point has been specified in procedure section.

In the present study, each interview was recorded, transcribed verbatim, and pro-cessed through the computerized IDAAP. The IDAAP was applied to the transcribed text separately for each topic that is neutral, positive, negative events and key-topic episode (fertility problems experience/pregnancy traumatic experience).

Through the one-way ANOVAs, the homogeneity of the two groups of participants was evaluated, specifically, for the sociodemographic continuous variables (normally distributed) and 296 discrete variables?

Authors: in the statistical section we reported both as follows

Through the one-way ANOVAs and the Chi-square tests the homogeneity of the two groups of participants was evaluated, respectively, for the sociodemographic continuous variables (normally distributed) and discrete variables

Why is the homogeneity of the groups checked?

Authors: we checked the homogeneity of the two groups for some variables in order to reduce the possibility that the difference found may be due to reasons different from the infertility/fertility experience. It is not possible to consider all possible variables but the simpler ones were evaluated.

What does table number three indicate?

Authors: table 3 offers clinical vignettes of narratives produced by women in the two groups that may be representative of the difference in the text produced.

We clarified in this sentence:

Table 3 reports some clinical examples of women’s narratives showing different emotional processes and engagement in the narrative task according to group belonging

  This report does not have the structure of a clinical article and cannot be followed due to clutter

Authors: we do not agree with this negative evaluation and we hope that the revised version, realized in accordance with the 3 reviwers’ suggestion may be improved in their quality and clearness and thus acceptable .

Round 2

Reviewer 2 Report

Comments and Suggestions for Authors

The authors are thanked for the changes they have made to the manuscript based on the suggestions offered by this reviewer. Although not every suggestion resulted in a change in the manuscript, the explanations provided by the authors for why the changes were not made were generally acceptable.

One thing the authors should keep in mind for the future is that, even when certain methods and procedures are the standard in a particular research area, the authors should think of why these methods and procedures continue to be appropriate right now. Science changes regularly. Therefore, ways of conducting research do as well. As such, this reviewer would prefer that the authors provide their reasons for using the methods they do, supported by current research in the area. Nevertheless, although this is the reviewer’s preference, the paper has been sufficiently modified to be up to the standard necessary for publication and the reviewer will not insist on further changes.

Author Response

Reviewer 2

The authors are thanked for the changes they have made to the manuscript based on the suggestions offered by this reviewer. Although not every suggestion resulted in a change in the manuscript, the explanations provided by the authors for why the changes were not made were generally acceptable.

Authors: We renew our thanks for the time dedicated to our manuscript and for the feedbacks provided. We are glade that you found the modified version satisfactory.

One thing the authors should keep in mind for the future is that, even when certain methods and procedures are the standard in a particular research area, the authors should think of why these methods and procedures continue to be appropriate right now. Science changes regularly. Therefore, ways of conducting research do as well. As such, this reviewer would prefer that the authors provide their reasons for using the methods they do, supported by current research in the area. Nevertheless, although this is the reviewer’s preference, the paper has been sufficiently modified to be up to the standard necessary for publication and the reviewer will not insist on

Authors: We are not sure of having correctly interpreted this point. We totally agree that methods procedures and theories should have a relevance in the current research to be valid and interesting for the readers.

If the issue is the general use of references older than 10 years vs more recent ones we think that we have integrated the pasted and the recent ones in line with your suggestion of using current research data. Recent findings, if in line with the less recent ones, can enrich an historical framework giving consistency to the work, in fact we added several recent papers from the link you suggested us.

If the point is the use of the Multiple Code Theory considered as overcame theory compared to others and so the associated research methodology considered as not actually relevant, we would like to report some consideration. MCT is a psychodynamic theory of the mind and of the allows to interpreted phenomenon from their double dimensions both physical and psychological so particularly useful for our purpose. Probably, and we agree with the reviewer, is not an easy theory and probably also for this it is not so common in the international papers. Nevertheless, this theory and the related methodology and procedure are still of interest as can be shown by recent papers we added on your request related to MCT applications in current research.

To prove that MCT is still valid and a theory of interest I can reported the link to a special issue of 2021 of the Journal of Psycholinguistic Research about Connecting Feelings, Thoughts, and Words: The Referential Process in Theory, Research, and Clinical Studies.

https://link.springer.com/journal/10936/volumes-and-issues/50-1

We added to the manuscript using a green font this two further sentences hoping to cleared the use of this theory and methods

One way to understand how people articulate emotionally difficult experiences, such as infertility, is through analyzing the way they speak, the language people use to tell the experiences faced. Indeed, language characteristics reflect the ability to connect emotions to words and thoughts [27]. In line with Bucci's Multiple Code Theory (MCT), that is a psychodynamic theory of the mind allowing to considered phenomenon in both their physical and mental dimensions, the linguistic analysis of spoken or written text enables the assessment of the connections between various methods or levels of processing and elaborating information [28–31]………. In highly traumatic or conflictual life events, related emotion patterns will also be more likely to generate dysregulated and difficult-to-manage arousal [31,38]. MCT so represent a valid and actual theoretical framework to explore how the words we use to construct a story related to our experience may reflect both emotional and cognitive elaborative capabilities of a dif-ficult/traumatic event.

Nevertheless, although this is the reviewer’s preference, the paper has been sufficiently modified to be up to the standard necessary for publication and the reviewer will not insist on

We appreciate your suggestion and that you considered sufficiently modified the papers.

Reviewer 3 Report

Comments and Suggestions for Authors

Many thanks for answers. 

Author Response

Many thanks for answers. 

Authors: we are glade that reviewer has found satisfactory our changes and the answer we provided. We renew our thanks for the time dedicated to our manuscript and for the feedback provided.